# Quantum phase synchronization via exciton-vibrational energy dissipation sustains long-lived coherence in photosynthetic antennas

Ruidan Zhu [1,5], Wenjun Li [2,3,5], Zhanghe Zhen[1,2,5], Jiading Zou [1,3], Guohong Liao [1,3], Jiayu Wang [1,3], Zhuan Wang [1], Hailong Chen [1,3,4], Song Qin [2,3] ✉ & Yuxiang Weng [1,3,4] ✉

The lifetime of electronic coherences found in photosynthetic antennas is known to be too short to match the energy transfer time, rendering the coherent energy transfer mechanism inactive. Exciton-vibrational coherence time in excitonic dimers which consist of two chromophores coupled by excitation transfer interaction, can however be much longer. Uncovering the mechanism for sustained coherences in a noisy biological environment is challenging, requiring the use of simpler model systems as proxies. Here, via two-dimensional electronic spectroscopy experiments, we present compelling evidence for longer exciton-vibrational coherence time in the allophycocyanin trimer, containing excitonic dimers, compared to isolated pigments. This is attributed to the quantum phase synchronization of the resonant vibrational collective modes of the dimer, where the anti-symmetric modes, coupled to excitonic states with fast dephasing, are dissipated. The decoupled symmetric counterparts are subject to slower energy dissipation. The resonant modes have a predicted nearly 50% reduction in the vibrational amplitudes, and almost zero amplitude in the corresponding dynamical Stokes shift spectrum compared to the isolated pigments. Our findings provide insights into the mechanisms for protecting coherences against the noisy environment.

Coherent energy transfer is a highly efficient energy transfer pathway in photosynthesis. Matching of long-lived quantum coherence to the time scale of energy transfer is a prerequisite[1–3]. In contrast to short-lived electronic coherence[4], the presence of exciton-vibrational coherence in photosynthetic systems[5,6] can account for the observed long-lasting quantum coherence. However, uncovering the mechanism of such coherence within a biological environment is challenging because of the presence of noise typically encountered at room temperature[7].

The optimized energy transfer (ET) process in photosynthesis occurring via light-harvesting pigment-protein complexes has a quantum efficiency approaching unity[8]. A quantum coherent ET mechanism has been proposed to account for this efficient ET process[1,2,9,10]. Coherent ET has obvious advantages over the classical energy hopping mechanism in that the former is able to employ interference of quantum amplitudes to steer ET, whereas the latter is constrained by classical probability laws[11]. Recently it has been shown that the wavelike quantum-coherent transfer channel is three times more efficient than the incoherent channel in a molecular dimeric model system[12]. This coherent ET mechanism can contribute to the near-perfect quantum efficiency of photosynthesis only if coherences

[1]Laboratory of Soft Matter Physics, Institute of Physics, Chinese Academy of Sciences, Beijing 100190, P.R. China. [2]Yantai Institute of Coastal Zone Research, Chinese Academy of Sciences, Yantai, Shandong 264003, P. R. China. [3]University of Chinese Academy of Sciences, Beijing 100049, P.R. China. [4]Songshan Lake Materials Laboratory, Dongguan 523808 Guangdong, P.R. China. [5]These authors contributed equally: Ruidan Zhu, Wenjun Li, Zhanghe Zhen. ✉e-mail: sqin@yic.ac.cn; yxweng@iphy.ac.cn

survive in these systems during ET at physiological temperatures. Although electronic and vibronic coherences exhibit delocalization, they are fragile at ambient temperature because of the rapid dephasing of electronic states[4].

In the past decades, there have been intensive investigations of quantum coherences in ET in the photosynthetic light-harvesting antenna known as the Fenna-Matthews-Olson (FMO) complex in the green sulfur bacterium *Chlorobaculum tepidum*[9,13–16]; this is because long-lived quantum beating signals of several hundred femtoseconds have been observed with two-dimensional electronic spectroscopy (2DES) at both cryogenic and room temperatures[9,13]. Various origins have been suggested for the observed coherences, namely electronic, vibrational, and vibronic coherences[17], though recent experimental evidence has suggested that the observed coherences are ground-state vibrational modes[14,18]. Currently, the leading hypothesis for coherence in ET is the quantum mechanical exchange of energy between excitonic and vibrational degrees of freedom[19], i.e., exciton-vibrational coupling[20], especially that between excitons and under-damped vibrations occurring in a resonant way[6,21–23], which would lead to the formation of delocalized joint quantum states[5,24]. However, a key question that remains unanswered is how these coherences can prevail over a long time in a physiological environment where there is aggressive fluctuation of protein vibrational modes and the surrounding polar solvents[25].

The long-lived coherences observed in photosynthetic antennas and the reaction center (RC), have been suggested to have a protective effect arising from the correlated movements of the protein matrix encapsulating the chromophores, where the slow protein motions cause synchronous fluctuations in the energies of the neighboring electronic states[26,27], hence rendering the delocalized electronic coherence less susceptible to environmental noise. Meanwhile, theoretical studies have focused on the other coherence-protection mechanisms, i.e., transient synchronization in a bio-inspired vibronic dimer, where electronic excitation dynamics are mediated by coherent interactions with intramolecular vibrational modes. The results of these studies have shown that coherent energy transport is concomitant with the emergence of positive synchronization (in phase) between mode displacements[19,23,28–31], which suggests that coherence in biomolecules promotes the synchronization of vibrational motions driven by thermal equilibrium or environmental noise. Transient spontaneous synchronization can lead to very weak energy dissipation due to collective motion[19,28]. Recently simulation work has shown

explicitly that the local underdamped vibrations in a dimeric pair, which are resonant with the electronic energy gap, are synchronized as the ET is completed[29]. Namely, the vibrations become anticorrelated (anti-symmetric collective mode) right after the initiation of ET and dissipate its energy to the environment through coupling to the delocalized electronic states. As ET is completed, the vibrations become synchronized (symmetric collective mode). The former mechanism assumes the correlated motion of the protein matrix, while the latter invokes the synchronous resonant vibrational displacement of the pigment pair. Yet experimental evidence for the protective effect of phase synchronization on long-lived coherences in a photosynthetic system during ET has not been reported.

Recombinant allophycocyanin (rAPC) trimer was used as an example to investigate this protective effect in excitonic dimers because it consists of three identical pairs of phycocyanobilin (PCB) molecules located on α- and β-subunits (Fig. 1a) respectively, which makes it an ideal model of a two-level system in photosynthesis. Allophycocyanin (APC) is an integral component of the light-harvesting antenna in blue and red algae, capable of delivering the energy absorbed from the rods of the phycobilisome to the RC with an overall quantum efficiency of over 90%[32]. The crystal structure of the APC trimer[33,34] revealed that it possesses a 3-fold axis of symmetry (Fig. 1a). The Cys84 sites in the α and β subunits of the monomer (α84 and β84, respectively) are covalently bound to a PCB pigment, forming a distance of ~50 Å. Upon trimerization, PCB pigments on different monomers are brought close enough together (~20 Å apart) to form three identical α84-β84 dimers. ET between the excitonically coupled PCB pigments has been reported to occur on a time scale of 200–400 fs[35–37], and intermolecular electronic couplings were suggested to be a critical factor for enhancement of the rate of ET in the APC trimer[30,36]. Thus, the APC trimer represents a simple dimeric system that is suitable to study the quantum coherence in ET and dissipation processes. We applied two-dimensional electronic spectroscopy (2DES) to study ultrafast ET, exciton-vibrational coherences, vibrational coherences and dynamical Stokes shift capable of reporting the coherent nuclear motion of the underdamped vibrational modes in the excited state[38,39] in the rAPC trimer and in the monomer and two respective α- and β-subunits without excitonic interaction for comparison.

Here we show evidence for long-lived exciton-vibrational coherence in the rAPC trimer containing excitonic dimers, compared to the isolated pigments. The coherence time in the trimer was extended to $501 \pm 104$ fs, compared to $93 \pm 19$ fs for the isolated pigments within

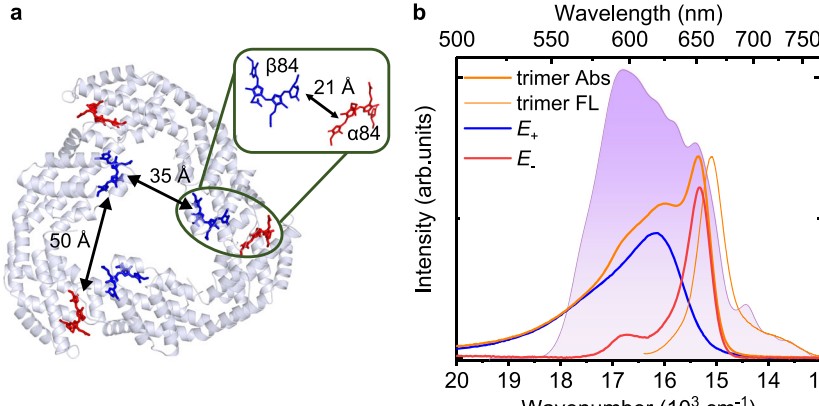

**Fig. 1 | The crystal structure of allophycocyanin (APC) and spectral characterization of recombinant allophycocyanin (rAPC). a** Crystal structure of APC with the β (blue) and α (red) phycocyanobilin pigments (PDB ID code 1ALL). In the trimer, a β-pigment is close to the α-pigment of a neighboring monomer with an electronic coupling of about 155 cm⁻¹. **b** The absorption (Abs, dark orange) and fluorescence (FL, light orange) spectra of the rAPC trimer are shown with the broadband spectral profile of the excitation laser pulses (shaded in violet). The absorption spectrum of the rAPC trimer is decomposed into two components corresponding to the upper exciton ($E_+$) and lower exciton ($E_-$) states[36]. The spectrum of the upper exciton (blue) is similar to that of the rAPC monomer, and the spectrum of the lower exciton (red) shows mirror-image symmetry with the fluorescence spectrum of the trimer. Source data are provided as a Source Data file.

the α-subunit. Quantum mechanical treatment of the resonant vibrational collective modes coupled to the excitonic levels reveals that, only the anti-symmetric modes can be coupled to the excitonic states of fast dephasing, while the decoupled symmetric counterparts of less energy dissipation survive, leading to the quantum phase synchronization of the resonant vibrational modes. Two indicators predicted by this theory were well confirmed by our experimental observations, i.e., the nearly 50% reduction in the vibrational amplitudes of the resonant modes and almost zero amplitude in the corresponding dynamical Stokes shift spectrum. Our results suggest that the quantum phase synchronization is responsible for the long-lasting quantum coherence and may play a role in the optimization of ET efficiency in light-harvesting complexes.

## Results and Discussion
### 2DES spectra and dynamical Stokes shift kinetics
To obtain high-purity monomers and two subunits without affecting their photophysical properties (Supplementary Fig. 5), we expressed the rAPC trimer in *Escherica coli*. The energy splitting ($\Delta E = 800\,cm^{-1}$) between the upper and lower excitons of the α84-β84 dimer in rAPC is significantly larger than the electronic coupling ($J = 155\,cm^{-1}$) as determined by circular dichroism spectroscopy (Supplementary Fig. 7), leading to a site energy ($\varepsilon$) difference of $\Delta_s = \varepsilon_{\beta84} - \varepsilon_{\alpha84} = \sqrt{\Delta E^2 - (2J)^2} = 737\,cm^{-1}$. In the 2DES experiment, a sequence of three ultrashort laser pulses excites the sample to induce a third-order polarization. During the time period between the first two pulses, the response evolves as a coherence between the ground and resonant excited states, and this initial time delay is also referred to as the coherence time ($t_1$). The second interaction promotes the evolution to populations of the excited states, populations of the ground state, and coherences between excited states. The time delay between the second and third pulses is also referred to as the population or waiting time ($t_2$). The emitted signal field is detected with a fourth pulse at $t_3$ after the last excitation pulse. The evolution of the excited states is then resolved in the 2D spectra relating excitation ($\omega_1$) and emission ($\omega_3$) frequencies, which are Fourier transform conjugates of the $t_1$ and $t_3$ time delays, respectively. Electronic or vibrational coherences are detected as oscillatory signals along the waiting time $t_2$, and are further Fourier-transformed to the corresponding coherence spectra[3,40,41]. The resonant ultrafast laser pulses with a broad spectral bandwidth create a coherent superposition of quantum states with a defined phase relationship across an ensemble, which in turn gives rise to quantum coherence beating signals. Dephasing of these quantum coherences is one of the most sensitive ways to probe the interactions between a system and its surrounding environment[17]. In particular, the dynamical Stokes shift is a sensitive means for probing the energy dissipation of the excited state as well as the dephasing of the vibrational coherence. Therefore, 2DES and dynamical Stokes shift detection provide potentiality for illustrating quantum phase synchronization of the resonant collective modes in the exciton dimer. In the rAPC trimer, the measurements of 2DES and dynamical Stokes shift quantify dephasing dynamics and provides clear evidence for the presence of quantum phase synchronization in the resonant collective modes, i.e., about half reduction in the vibrational amplitudes of the resonant modes in the coherence spectra and their almost zero amplitude in dynamical Stokes shift spectrum. Our results suggest that the quantum phase synchronization is responsible for the long-lasting quantum coherence and play a role in the optimization of ET efficiency in light-harvesting complexes.

In the measured 2D spectra of the rAPC trimer and α-subunit, two diagonal peaks were observed for the trimer, with positive values corresponding to the bleaching/emission of the upper and lower excitonic levels (Fig. 2a and Supplementary Fig. 10), while only a single elongated diagonal peak was observed for the α-subunit (Fig. 2b). The 2D spectra of the monomer and β-subunit show the same feature as

that of α-subunit (Supplementary Fig. 9). The peaks below the diagonal with negative amplitude are the excited-state absorption peaks. The possible compensation effect of the excited-state absorption signals for the kinetic analysis of the monomer and two subunits is excluded as discussed in Supplementary Fig. 13. Over time there is a redshift of diagonal peaks along the emission frequency (i.e., dynamical Stokes shift) and spectral broadening of diagonal peaks along the anti-diagonal direction (i.e., spectral diffusion); this reflects the interaction of the chromophores with the surrounding protein matrix and solvent molecules, a dynamical process known as solvation[42]. Meanwhile, ET between the two excitonic levels in the trimer is clearly shown by the increased amplitude of the cross peak below the diagonal ($16,000\,cm^{-1}$, $15,300\,cm^{-1}$) and the concomitant decrease of the diagonal peak ($16,000\,cm^{-1}$, $16,000\,cm^{-1}$) for the upper excitonic state in the 2D maps over time[16] (Fig. 2c). The two diagonal peaks and one cross peak are clearly resolved in the corresponding 2D maps acquired at 77 K (Supplementary Fig. 11). The ET lifetime, 220 fs for the decay envelope, extracted using the global fitting procedure (Supplementary Fig. 12) is consistent with that observed in previous studies of natural APC trimers[35,37]. In contrast, the spectral evolution of the α-subunit (Fig. 2b) is dominated by the contributions of dynamical Stokes shift and spectral diffusion of the isolated PCB in the protein matrix. The decay kinetics of its excited state can be fitted by three-exponential decay, with the fastest lifetime constant of 140 fs. Pronounced quantum coherence signals were observed for both the trimer and α-subunit after subtraction of the fitted decay envelopes from the acquired population decay kinetics (Figs. 2d and e). For the trimer, quantum coherence can arise from either exciton-vibrational coupling or vibration, while for the α-subunit only vibrational coherence is possible. This is due to that the α/β subunit contains only one pigment in the protein. In addition, the rAPC monomer contains two pigments, which are spatially separated by a distance of 50 Å. Consequently, the intermolecular electronic coupling between these two pigments is negligible. As a result, the coherent oscillating signal observed in the monomer and two subunits can solely be attributed to that of intramolecular vibration.

In a 2DES experiment, the system is excited with ultrafast laser pulses. This impulsive laser excitation of vibrational modes to the Franck-Condon region in the molecular excited state produces wavepackets in the excited state and "holes" in the ground state. As time evolves, the excited state begins to relax through intramolecular and intermolecular vibrational relaxation and solvent reorganization. Relaxation of the excited electronic state manifests as a shift in frequency along the emission frequency of the 2DES spectrum as the energy gap between the ground and excited state decreases. This energy relaxation is referred to as the dynamic Stokes shift and can be described by the Stokes shift function $S(t)$, which exhibits oscillatory behavior, and the oscillatory components are assigned to intramolecular vibrational modes. Fourier transformation of $S(t)$ gives rise to the coherence spectrum of dynamical Stokes shift. Therefore, the dynamic Stokes shift is a common means for characterizing ultrafast solvation dynamics of electronically excited states, hence the energy dissipation to the environments as well as the intramolecular or solvent vibrational modes participating in the energy dissipation[39,43,44]. Energy dissipation dynamics in terms of the dynamical Stokes shift function $S(t)$ were extracted from the 2D spectra by monitoring the shift of the diagonal peak along emission frequency as a function of time for the lower excitonic level of the trimer at $\omega_1 = 15,300\,cm^{-1}$ and for the diagonal peak of the α-subunit at $16000\,cm^{-1}$ (Fig. 3 and Supplementary Fig. 14). The dynamical Stokes shift $S(t)$ can be well fitted with the sum of a Gaussian and damped sine function $f(t) = A_d \exp[-t^2/(2\tau_d^2)] + A_c \exp(-t/\tau_c)\sin(2\pi\omega \cdot t + \phi) + C$, where $\tau_d$ and $\tau_c$ are the decay and coherent lifetimes, respectively, $A_d$ and $A_c$ are the corresponding amplitudes, $\omega$ ($\phi$) is the oscillation frequency (phase), and $C$ is the constant term. The energy relaxation dynamics, shown as the envelope with a decay lifetime ($\tau_d$) of about

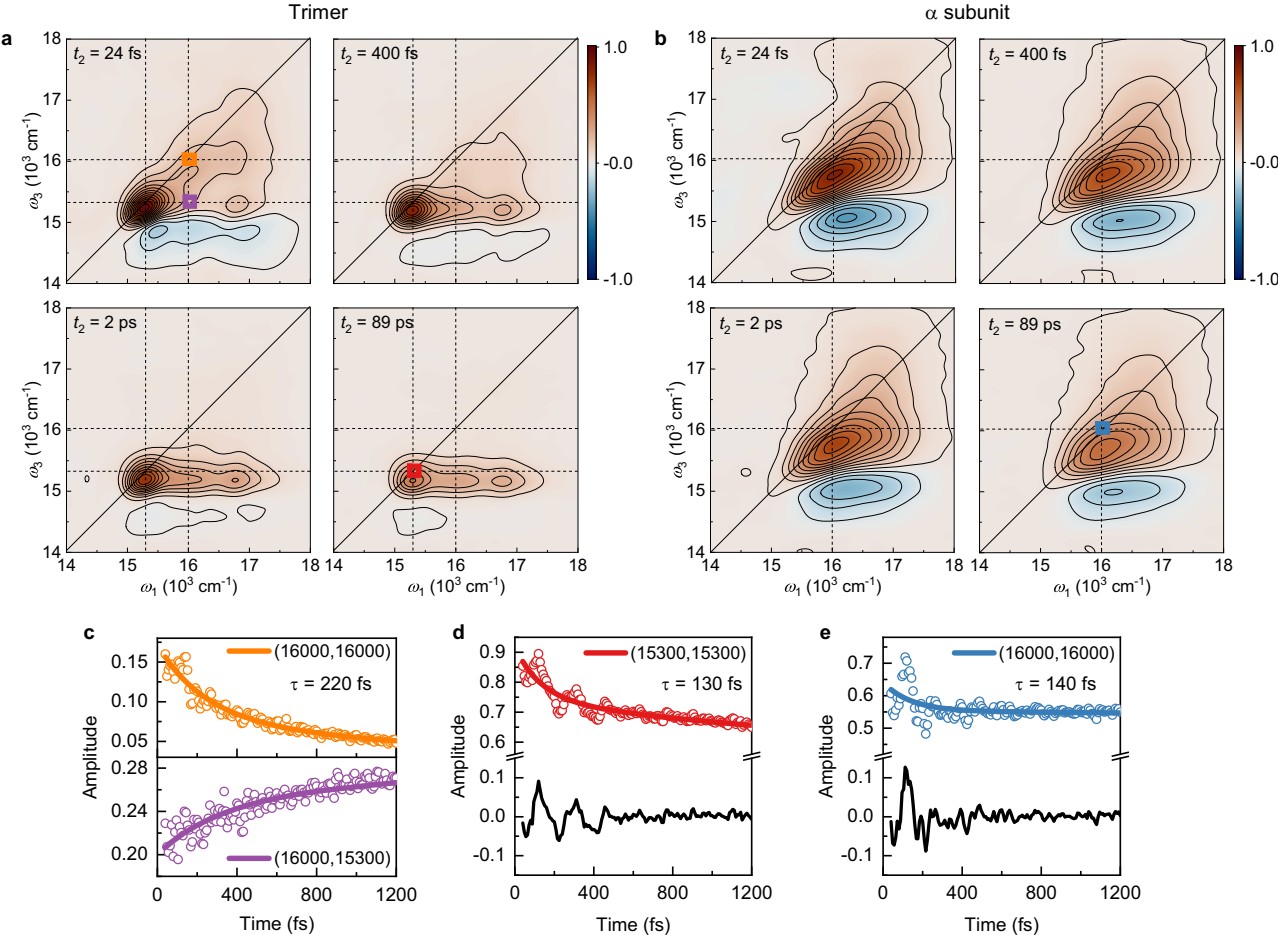

**Fig. 2 | Comparison of time-dependent 2D spectra between the rAPC trimer and α-subunit at room temperature. a** Absorptive 2D spectra of the rAPC trimer over time with dashed lines indicating the upper and lower excitonic energy levels. Contour lines are drawn in 7.6% intervals. **b** Absorptive 2D spectra of the α-subunit with dashed lines indicating the electronic transition energy. Contour lines are drawn in 7.8% intervals. **c** The energy transfer kinetics of the trimer at the upper diagonal (top) and the cross peak (bottom) respectively. **d** The energy relaxation kinetics of the trimer at the lower diagonal peak. **e** The energy relaxation kinetics of the α-subunit at the diagonal peak. The global fits (lines) with a four-exponential

function $f(t) = \sum_{i=1}^{4} A_i \exp(-t/\tau_i)$ are superimposed on the raw data (dots), where $\tau$ is lifetime constant, and $A$ is the corresponding coefficient. Only the fastest life-time constants are shown, and other lifetime constants can be found in Supplementary Fig. 12. The numbers and colors in the legend of (**c–e**) indicate the positions of the peaks corresponding to these kinetics in the 2D spectra, which are marked with squares in the same color as in **a** and **b**. The residues (black lines) show large coherent amplitudes up to 10% on the diagonal peaks relative to the max-imum population signal. Source data are provided as a Source Data file.

100 fs, were comparable for the trimer, monomer and two subunits (Supplementary Table 1). This continuous relaxation of energy corre-sponds to the inertial solvation response of the protein and solvent[45]. However, we observed that the coherence lifetime ($\tau_c$) was significantly shorter for the α- subunit ($93 \pm 19$ fs) compared with the trimer ($501 \pm 104$ fs). These oscillations in $S(t)$ originate from the coherent coupling of vibrational modes to electronic states as a wavepacket[43,46]. This suggests that the long-lived coherence in the trimer is strongly associated with the exciton-vibrational coupling in the α84-β84 dimer. Additional evidence for the prolongation of coherence can be found in the broadband transient absorption (BBTA) measurement of the dyna-mical Stokes shift function (Supplementary Table 2).

**Coherence spectra from population beating and dynamical Stokes shift**
To reveal the dominant coherence beating frequencies that are present in 2D spectra, we projected the 2D spectra along the exci-tation frequency, then performed Fourier transformation along the waiting time $t_2$ on the oscillatory residual, and finally integrated the coherent amplitude along the emission frequency for the trimer and α-subunit. Eleven significant coherent frequencies were observed

(gray lines in Fig. 4) in the coherent spectra for both the trimer and the α-subunit. The beating signals in APC trimer are in agreement with those observed by transient absorption spectroscopy with similar vibrational frequencies up to 1000 cm$^{-1}$ as the limit in their measurement[37]. These frequencies are also consistent with the reported vibrational frequencies of PCB molecules[47,48]. In particular, two vibrational modes in the region of near-resonance with the excitonic energy splitting (orange-shaded area) were assigned to ethylenic hydrogen out-of-plane (HOOP) wagging (805 cm$^{-1}$) and mixed HOOP bending, methine torsions, and ring rotations (660 cm$^{-1}$); these HOOP motions could affect the inter-pigment distance. Assignments of the other vibrational modes are summarized in Supplementary Table 3. A comparison of the integrated coherent spectrum of the trimer with that of the α-subunit (top of Fig. 4a) revealed an approximately 50% reduction (Table 1) in the amplitudes at the two near-resonant frequencies (the averaged reduction ratio for trimer with respect to the monomer and two different subunits over three different kinds of measurements gives rise to a value of $49\% \pm 7\%$ in Table 1). This spectral feature was reproducible in additional BBTA and heterodyne-detected transient grating (HDTG) measurements (bottom of Fig. 4a and Supplementary Figs. 15-18).

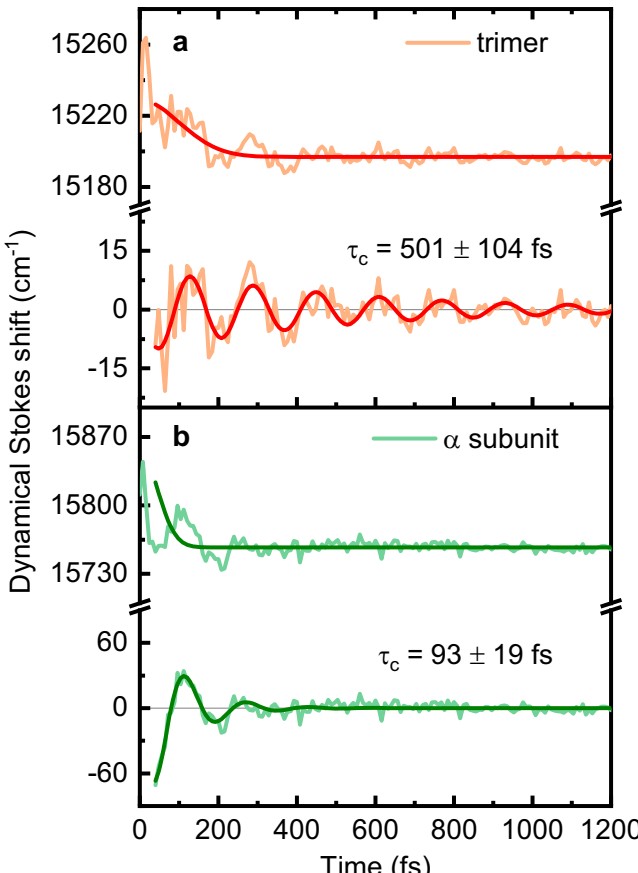

**Fig. 3 | The long-lived coherence in the dynamical Stokes shift of the rAPC trimer.** The dynamical Stokes shift of the diagonal peaks at 15300 cm$^{-1}$ in the lower excitonic level of the trimer (orange line in **a**) and at 16000 cm$^{-1}$ in the α-subunit (light green line in **b**) are detected via two-dimensional electronic spectroscopy (2DES). The energy relaxation kinetics are fitted with the sum of a Gaussian and damped sine function mentioned in the main text. The fitting results (red lines in **a** and dark green lines in **b**) comprising the Gaussian decay component and the oscillation component are superimposed on the original kinetics (top) and oscillatory residual curves (bottom) for each panel. The damped oscillation decay ($\tau_c$) time constants indicated on the curves reveal the dephasing of coherence. Source data are provided as a Source Data file.

We found that such a reduction of amplitudes at the resonant frequencies does not occur in the non-resonant Raman spectrum of the trimer with respect to that of the monomer (Supplementary Fig. 8). We noted a reduction of amplitude at 270 cm$^{-1}$ in the low frequency and non-resonant regions for the trimer arising from a reduction in the corresponding Huang-Rhys factor; this phenomenon also appears in the non-resonant Raman spectrum of the trimer because of a change in the local structure of the PCB after trimerization (Supplementary Fig. 8). It has been reported that formation of dimers reduces the coupling with vibrational degrees of freedom in the photosynthetic complexes, leading to the low-frequencies in a region below 300 cm$^{-1}$ being strongly suppressed by excitonic/vibronic delocalization, while the high-frequency modes are likely to be unaffected[49,50]. Therefore, the observed about 50 % reduction in the intensities of the vibrational modes between 600-820 cm$^{-1}$ cannot be caused by the excitonic delocalization in the dimer.

Coherent spectra of dynamical Stokes shift were shown in Fig. 4b, and these spectra showed that the resonant modes almost completely disappeared, which indicates that these resonant modes do not participate in the energy dissipation of the lower excitonic level of the trimer at $\omega_1 = 15,300$ cm$^{-1}$.

## Accounts for the experimental observations by quantum phase synchronization

To account for the observed long-lived coherence, about 50% reduction in the amplitudes of the resonant modes in the coherence spectrum, and disappearance of the resonant modes in the coherent spectrum of dynamical Stokes shift for the trimer, we exploited the effect of phase synchronization on 2DES in the exciton-vibrational model of the dimer, in terms of the symmetric/anti-symmetric collective vibrational basis (Supplementary Note 11). This phase synchronization idea was inspired by the classical analogue known as Huygens' clock (Fig. 5a), where the two pendulums are coupled. The in-phase collective motion is more energy dissipative than the anti-phase one, resulting in the anti-synchronized motion of the pendulum clocks[51]. If this phase synchronization mechanism also presents in the exciton-vibrational coupled dimer, it can be expected that phase synchronization of the resonant collective modes would lead to these modes becoming less energy dissipative as indicated by our experimental observation in Fig. 3.

As illustrated in Fig. 5, the model consists of a non-degenerate homodimer (with the monomers designated as A and B). Each molecule has an identical intramolecular harmonic vibration with a frequency of $\omega$ coupled to the electronic states. The corresponding Hamiltonian, including the ground state and two singly excited states (excitonic levels), is given by

$$\hat{H} = T + [V_A(q_A) + V_B(q_B)]|g\rangle\langle g|$$
$$+ [\varepsilon_A + V_A(q_A^*) + V_B(q_B)]|e_1\rangle\langle e_1|$$
$$+ [\varepsilon_B + V_A(q_A) + V_B(q_B^*)]|e_2\rangle\langle e_2|$$
$$+ J[|e_1\rangle\langle e_2| + |e_2\rangle\langle e_1|] \tag{1}$$

where $T$ is the kinetic energy; $V(q)$ is the potential energy surface; $q^*$ and $q$ are the dimensionless coordinates of the excited and ground states for the two molecules, respectively; and $|g\rangle$ and $|e\rangle$ represent the ground and excited states on a local basis, respectively. We then convert the intramolecular vibrational modes into the respective intermolecular collective modes, i.e., $Q_s = (q_A + q_B)/\sqrt{2}$ is the symmetric collective mode and $Q_a = (q_A - q_B)/\sqrt{2}$ is the anti-symmetric collective mode[6]. Under weak electronic coupling and a significant site energy difference, i.e., $\Delta_s/2 \gg J$, the eigen energies of the two delocalized states can be derived as follows:

$$E_{\pm} = \frac{\varepsilon_A + \varepsilon_B}{2} + \frac{\hbar\omega\sqrt{S}}{2}[Q_s^* + Q_s] \pm \left[\frac{\Delta_s}{2} + \frac{\hbar\omega\sqrt{S}}{2}(Q_a^* + Q_a)\right] \tag{2}$$

where the subscript $\pm$ indicates the upper and lower energy levels.

After impulsive optical excitation, the intramolecular vibrational mode experiences a Brownian force from the environment and undergoes underdamped oscillation[38], and the excited states undergo energy dissipation. Hence the dynamical Stokes shift reflects a red shift in $E_{\pm}$ over time. By deriving the underdamped motion equation of the coordinate $q(t)$ in the form of oscillation with a frequency of $\omega$ and an initial phase of $\phi$, the time-dependent energy levels $E_{\pm}(t)$ can be expressed as a function of $Q(t)$.

Consequently, the time-dependent dynamical Stokes shift of the lower excitonic energy level in a weakly coupled excitonic dimer with a large site energy difference (Supplementary Note 11), which can be determined experimentally, is given as follows:

$$\Delta E_{-}(t) = \sum_{i=e,g} A_0^i \sqrt{S}\hbar\omega\, e^{\frac{\gamma t}{2}} \sin\left(\frac{\varphi_s^i - \varphi_a^i}{2}\right)\left[-\sin\left(\omega t + \frac{\varphi_s^i + \varphi_a^i}{2}\right)\right.$$
$$\left. + \frac{\gamma}{2\omega}\cos\left(\omega t + \frac{\varphi_s^i + \varphi_a^i}{2}\right)\right] \tag{3}$$

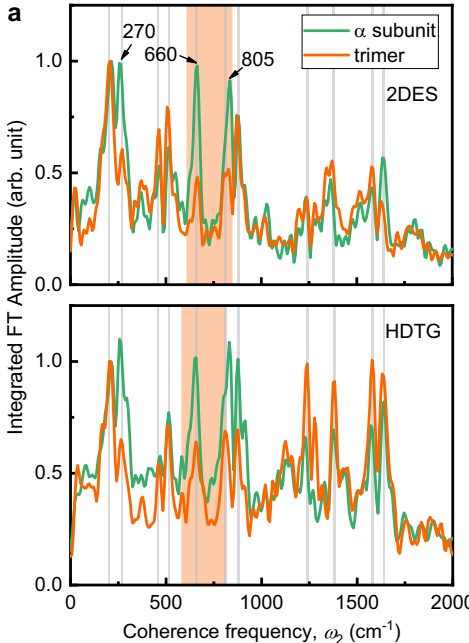

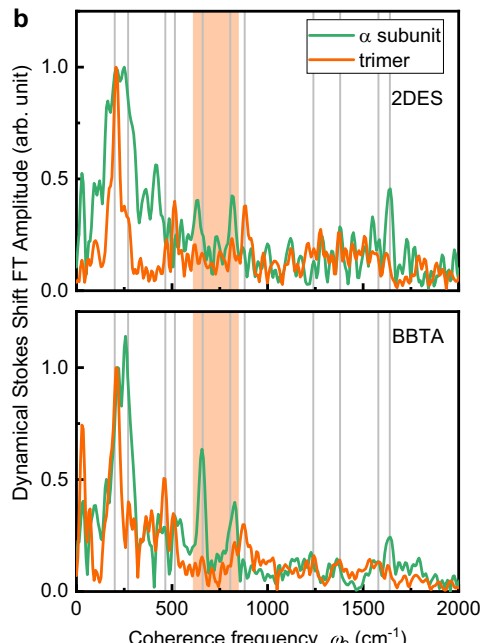

**Fig. 4 | Reduction in amplitudes of resonant vibrational modes in the coherence spectra and absence of the resonant mode in the dynamical Stokes shift spectra of the rAPC trimer. a** Comparison of the integrated coherence spectra between the α-subunit (green) and trimer (orange) from 2DES (top) and heterodyne-detected transient grating (HDTG, bottom) measurements. HDTG measurements were obtained with a broader excitation spectrum. The background noise level of 2DES(HDTG)-detected coherence spectrum was about 15% (19%) of the integrated FT spectrum maximum. **b** Coherence spectra of dynamical Stokes shift measured via 2DES (top) and broadband transient absorption (BBTA, bottom) with the same excitation spectrum. The background noise level of 2DES(BBTA)-detected coherence spectrum was about 12% (4%) of the FT spectrum maximum. The gray lines indicate vibrational modes observed in both spectra: 200, 270, 465, 515, 660, 805, 880, 1240, 1380, 1580, and 1640 cm⁻¹. The orange-shaded area shows the near-resonant frequency region in the trimer. All spectra were normalized to the amplitude at 200 cm⁻¹. Source data are provided as a Source Data file.

where $A_0$, $\gamma$ and $S$ are the oscillation amplitude, damping rate of the collective modes, and the Huang-Rhys factor of the given intramolecular mode, respectively. Equation (3) shows how the initial phases (designated as the superscript $i$) of the symmetric and anti-symmetric collective modes of the excited and ground states affect the dynamical Stokes shift of the lower excitonic state. In general, in a weakly coupled dimer these two molecules behave almost independently, giving rise to a random initial phase ($\varphi_a^i$ and $\varphi_s^i$) for the two collective modes[28]. Hence the motion of these two collective modes is uncorrelated, having no difference in ensemble average.

Next, we further explored the possibility of phase synchronization in this exciton-vibrational coupling system after impulsive excitation. We changed the local electronic wave function basis to the delocalized excitonic basis using the Hamiltonian in the Jaynes-Cummings form[28] (Supplementary Note 11) and obtained the exciton-vibrational coupling term $H_{coupling} = (1/2)\hbar\omega\sqrt{S} \sin(2\theta)(Q_a^* + Q_a)\sigma_x$, where $\theta$ is the mixing angle representing the extent of electronic delocalization in the dimer, and $\sigma_x$ is the transition operator. Only the anti-symmetric collective modes thus can be coupled to the two delocalized excitonic states. Therefore, once the anti-symmetric collective modes are in resonance with the two delocalized excitonic states, vibrational energy will be dissipated to the environment through fast excitonic dephasing[35] (30 fs⁻¹) owing to wave function mixing between the electronic and vibronic states[6], leaving only the symmetric collective modes survived. This automatically leads to the synchronization of the two collective modes after a certain period of time, even though the initial phases can be different, i.e., the initial phase of the symmetric and anti-symmetric collective modes become the same ($\phi_s - \phi_a = 0$), and the symmetric and the anti-symmetric collective modes become phase correlated with a fixed difference of $\pi$. This provides a mechanism for the exciton-vibrational coupling-initiated resonant vibrational phase synchronization in an excitonic dimer. Since the energy of the resonant anti-symmetric modes is dissipated while the symmetric modes remain unaffected, this gives rise to a loss of half of the intensity of the resonant modes in the coherence spectra, which is consistent with the experimental observations in Fig. 4a and Table 1. Furthermore, the initial phase synchronization would lead to the disappearance of the resonant modes in the dynamical Stokes shift spectrum as predicted by Eq. (3), i.e., $\Delta E_-(t) = 0$ at the given resonant vibrational mode, which is also consistent with the results presented in Fig. 4b. Therefore, the two predictions are fully in agreement with the experimental observations. Furthermore, these two predicted and experimentally proved features can serve as criteria for the identification of resonant exciton-vibrational coherence in a weakly coupled excitonic dimer with a large site energy difference. Because the motion of the symmetric collective modes does not alter the separation distance in the dimer, and hence its electronic coupling, and because the symmetric motion is much less energy dissipative than the anti-symmetric motion, which changes the intermolecular separation, the synchronous vibrational motion of the two pigments in the dimer is robust to protein fluctuations. This would have a protective effect on

**Table 1 | Reduction ratio in amplitudes of resonant vibrational modes in the integrated coherence spectra**

| | Ratio ($\frac{Trimer}{\alpha-subunit}$) | Ratio ($\frac{Trimer}{\beta-subunit}$) | Ratio ($\frac{Trimer}{Monomer}$) |
|---|---|---|---|
| 2DES | 51% | 41% | 43% |
| HDTG | 55% | 59% | 51% |
| BBTA | 55% | 44% | 41% |

The reduction ratio in amplitudes of the trimer is obtained by dividing the average FT amplitude of the trimer above the noise level in the orange-shaded region in Fig. 4a by the average FT amplitudes of the monomer and two subunits in the same region. Given the equivalence of three measurements (2DES, HDTG, and BBTA), these nine independent results yielded an average reduction ratio of 49% ± 7%.

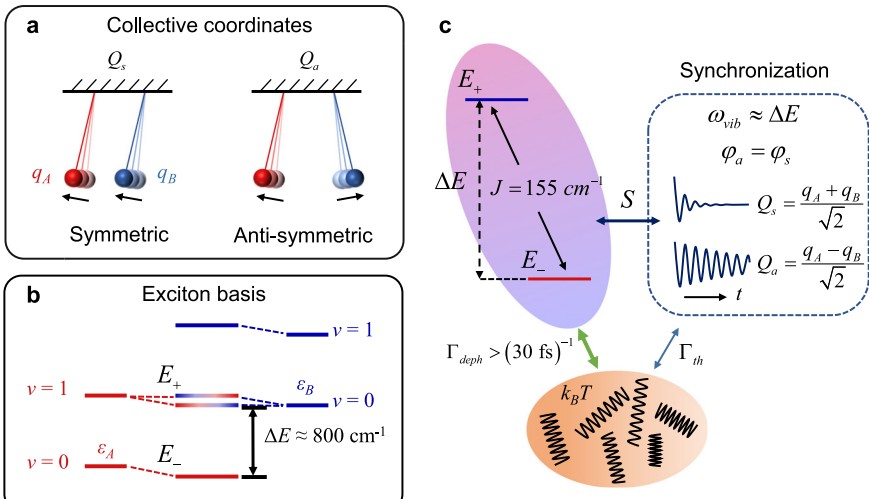

**Fig. 5 | Mechanism of quantum phase synchronization. a** Schematic diagram of symmetric ($Q_s$) and anti-symmetric ($Q_a$) collective coordinates with reference to a pair of Huygens pendulums. $q_A$ and $q_B$ are localized intramolecular vibrational coordinates. **b** The energy diagram of the exciton-vibrational dimer. $E_+$ and $E_-$ are the eigen energies of the two delocalized states under the exciton basis. **c** The mechanism of energy dissipation by the anti-symmetric modes via coupling to the delocalized excitonic levels with a rate constant of $\Gamma_{deph}$; the symmetric modes undergo dephasing by coupling to the thermal bath with a rate constant of $\Gamma_{th}$.

the long-lasting exciton-vibrational coherence of the trimer as shown in Fig. 3.

Another spectroscopic feature in 2DES has been suggested to represent the exciton-vibrational coherence in molecular systems, i.e., the asymmetric amplitude enhancement at the cross peak below the diagonal in the rephasing coherence maps is an indication of the existence of exciton-vibrational coherence. This is because the non-adiabatic vibrational-electronic mixing resonantly enhances the amplitude of delocalized, anticorrelated vibrational wave-packet motion in the ground electronic state[6]. We also observed an asymmetric enhancement of amplitudes of the 660 and 805 cm[-1] peaks in the rephasing coherence maps for the rAPC trimer (Supplementary Fig. 19). However, a similar asymmetric enhancement of amplitude also occurs in the coherence map of the α-subunit. Since there is only pure vibrational coherence in the α-subunit, this indicator fails to identify the resonant exciton-vibrational coherence in the rAPC trimer. The reason for the amplitude enhancement at the cross peak below the diagonal in the rephasing coherence maps for both the trimer and monomer could be that, in the vibrational coherence map, the ground-state vibrational coherence is dominant over that of the excited-state as revealed by the corresponding Feynman diagrams[52,53].

We noted that Christensson et al., explored theoretically the origin of long-lived coherences in FMO using a vibronic exciton model. They demonstrated that a biphasic decay feature exists in the vibronic exciton model, explicitly, the oscillations show a biphasic behavior, where the initial 200 fs decay of the oscillation is due to the decay of coherences between vibronic exciton states localized on different pigments, while the long-lived oscillations reflect coherences between vibronic exciton states localized on the same pigment (intrapigment) due to strong exciton-vibrational coupling at the near-resonant condition with an excitonic energy difference $\Delta E = 217$ cm[-1] and a vibrational frequency $\omega = 185$ cm[-1]. where the corresponding excitonic state contains 68% of the vibrational mode manifesting a strong exciton-vibrational coupling. This leads to that the two exciton levels will experience highly correlated fluctuations. Consequently, they explained the long-lived coherences based on coherent superpositions of vibronic exciton states with dominant contributions from vibrational excitations on the same pigment. This type of correlation leads to slow homogeneous dephasing of coherences with large intrapigment characters. Apparently, this correlation is an inherit feature of a single molecule but not of two individual molecules in the dimer of

weak coupling. In the current work, we also observed the biphasic dephasing of the vibronic exciton, the 10-30 fs fast decay of APC trimer is from pure electronic coherence[35]. While the slow decay phase has a strong vibronic feature, i.e., the wavefunction is a mixture of the electron and nuclear motion through Franck-Condon transition. The electronic coupling strength $|\frac{V}{\Delta E}| = 0.42$ is rather weak, where$\Delta E$ is the disorder of the site energy, $V = 2|J|$ is the excitonic coupling matric element. Furthermore, the Huang-Rhys factors for the vibrational modes larger than 490 cm[-1] for the APC trimer are small (less than 0.1)[37]. Both the facts suggest that APC trimer is a weak electronic and excitonic-vibrational coupling system. Owing to the large site energy, the two exciton levels are located almost on different pigments, thus the two excitonic levels cannot be considered as of intrapigment character. However, once the anti-symmetric collective modes have been dissipated through electronic dephasing owing to their coupling to the excitonic levels, only the total symmetric collective modes remain. This leads to the energy fluctuation of the two excitonic levels in a synchronous correlation, and the correlated pigment pair in the dimer would behave like a single molecule, having the intrapigment feature. Unlike the Christensson's vibronic exciton model, the seemingly intrapigment feature of the weakly coupled dimer is not an inherent property, but it is an acquired one by phase synchronization, and it takes some period of time for the realization of phase synchronization. As a result, Christensson's vibronic exciton model describes the long-lived coherences arising from strong exciton-vibrational coupling with intrapigment feature, while the current quantum phase synchronization mechanism is valid for weak exciton-vibrational coupling system.

We found that the coherence lifetime of rAPC ranges from ~100 fs for the α-subunit to ~500 fs for the trimer, suggesting that the exciton-vibrational coherence of the excitonic dimer in the rAPC trimer is protected against fluctuating environmental noise. Furthermore, in the trimer, the acquired coherent vibrational spectra revealed a ~50% reduction in the amplitudes of two resonant vibrational modes, consistent with the prediction that there is a loss of energy from the anti-symmetric resonant collective vibrational modes to the environment. In contrast, the resonant vibrational frequency was almost absent in the dynamical Stokes shift spectra, illustrating that the symmetric resonant collective modes are much less active in the energy dissipation of the excited states. These two features are consistent with the predictions of quantum phase synchronization of weak electronic coupling.

The quantum phase synchronization in the excitonic dimer provides a mechanism for the emergence of long-lived exciton-vibrational quantum coherence in a biological system at room temperature. Our work indicates that by harnessing the quantum feature of exciton-vibrational coupling, the dimer, which serves as a light-harvesting unit, resists the loss of excitation energy to the environment and hence achieves a high ET efficiency. It also provides an important mechanism of sustaining long-lived coherences in quantum biology.

## Methods

### Sample preparation

The APC trimer from the cyanobacterium *Synechocystis* sp. PCC 6803 was prepared and purified from *Escherichia coli* as described previously[54]. The monomer was prepared by disassembling the rAPC trimer using 300 mM imidazole with 20 mM phosphate, followed by separation using sucrose density gradient centrifugation. Two sub-units were prepared by the heterologous recombinant expression with a slight modification of the double plasmid. The detailed procedures, plasmid information, and methods for protein identification are described in Supplementary Note 1 and Supplementary Figs. 1-4. For ultrafast spectroscopy experiments, the rAPC trimer was dissolved in 750 mM phosphate buffer, and the monomer was dissolved in 20 mM phosphate with 300 mM imidazole to prevent trimerization. Two subunits were dissolved in 20 mM phosphate buffer. The solutions for 2DES measurements of the rAPC trimer and α-subunit were prepared at room temperature with average optical densities of ~0.5 and ~0.35, respectively, at the maximum absorption in a 1-mm path length cuvette. The sample's integrity was confirmed by comparing the linear absorption spectra before and after the experiment.

### Two-dimensional electronic spectroscopy

2D spectra were measured using a fully non-collinear BOXCARS 2DES setup described previously[55]. A home-built non-collinear optical parametric amplifier pumped by a Ti:Sapphire laser (Spitfire Ace, Spectra-Physics) with a 1-kHz repetition rate generated broadband laser pulses spanning from 550 to 720 nm. The pulses were compressed to ~8 fs (Supplementary Fig. 6) using the combination of a grating pair and a fused silica prism pair[56]. In the 2DES setup, the pulse was split into four phase-locked pulses using two beam splitters (Layertec). All four pulses are of parallel polarization. One of the pulses was attenuated by about three orders of magnitude and used as a local oscillator for heterodyne detection. Four phase-locked pulses were focused on the sample to emit the third-order 2DES signal along the phase-matching direction. The heterodyne interference signal was detected by an Acton SP2358 spectrometer (Princeton Instruments) and an EMCCD camera (ProEm 1600$^2$, Princeton Instruments). 2D spectra of the rAPC trimer for each time point were collected by scanning the coherence time $t_1$ from -67.5 fs to 67.5 fs in a 1.35-fs step. For the rAPC monomer and two subunits, the coherence time was scanned from -40.5 fs to 40.5 fs with a 1.35-fs step. The waiting time $t_2$ was linearly scanned up to 1.2 ps in an 8-fs step and then non-linearly scanned up to 94.4 ps. An excitation energy of 4 nJ per pulse was used for all 2D measurements, which corresponds to an excitation energy density of 20 μJ/cm$^2$, which is in the linear response region of samples. The 2D data of the trimer and α subunit were averaged seven and six times, respectively, to ensure reproducibility. Fourier transformation along $t_2$ was performed from 40 fs to 1.2 ps to eliminate the overlapping pulse effect in the coherence analysis. Thus, the actual frequency resolution along $\omega_2$ was ~28 cm$^{-1}$, while the displayed $\omega_2$ resolution for all FT spectra was set to 7 cm$^{-1}$ after zero-padding. In parallel-polarized HD-TG experiments, the time delay between the first two pulses was fixed at zero. The waiting time was scanned from -40 fs to 1.2 ps in an 8-fs step. The fluence of the excitation pulses was the same as that in the 2D measurement. The HD-TG spectra were phased using the projection slice theorem to match

the BBTA results under the same conditions. In parallel-polarized BBTA experiments, the first and third pulses were blocked in the 2D setup, and we used the second and fourth pulses as the pump and probe pulses, respectively.

Detailed descriptions of pulse characterization and HDTG/BBTA spectroscopy are provided in Supplementary Notes 1 and 8, respectively.

## Data availability

The APC structure used in this study were obtained from the Protein Data Bank (PDB) with accession codes 1ALL. All experimental data generated in this study are provided as a Source Data file with this paper. Source data are provided with this paper.

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

## Acknowledgements

This work was supported by the Natural Science Foundation of China (T2350011 to Y.W., 22027802 to Y.W., U2230203 to Z. W. and 41906109 to W.L.), and the Strategic Priority Research Program (XDB33000000 to Y.W.), Natural Science Foundation of Shandong Province (ZR2021LLZ003 to Y.W and W.L.). We thank Prof. Junpeng Cao and Prof. Wei Zhuang from the Chinese Academy of Sciences for helpful discussion about the theoretical sections and proofreading.

## Author contributions

Y.W. and S.Q. conceived and supervised the project. Z.Z. and W.L. prepared and characterized the samples with steady-state spectroscopy. R.Z. and J.Z. performed ultrafast spectroscopic experiments. R.Z., Y.W. and J.W. analyzed the experimental data. Y.W. and G.L. formulated the theoretical description of the model. All authors, including H. C. and Z. W. discussed the results. Y.W., R.Z., W.L. and Z.Z. wrote the manuscript with input from all co-authors. R.Z. and Z.Z. prepared all figures.

## Competing interests

The authors declare no competing interests.
