## [Peer Review File · Nature Communications]

Quantum phase synchronization via exciton-vibrational energy dissipation sustains long-lived coherence in photosynthetic antennasReviewers' comments:

Reviewer #1 (Remarks to the Author):

The authors address the question how long-lasting coherences can survive in the presence of a noisy environment as it is the case for most biological systems. Using two-dimensional spectroscopy as well as other techniques, they probe the coherences in allophycocaynin trimers and monomers. The authors explain the differences of the monomer and trimer measurements with the help of a vibronic dimer model and identify two spectroscopic signatures of phase synchronization. The authors conclude that phase synchronization is a mechanism that explains long-lasting coherences in biological systems.

Vibronic coherences in photosynthetic antenna complexes are an ongoing research topic and thus the study is of fundamental scientific significance. The analysis is in principle convincing, and I recommend this work for publication in Nature Communications. However, I have a few issues that need to be addressed prior to publication.

Comments to the main manuscript:

Line 63: The term "excitonic-vibronic coupling" is defined as the "...exchange of energy between excitonic and vibrational degrees of freedom". The term "excitonic-vibronic coupling" is in my opinion confusing since in literature [compare, for example, ref. 19, Novelli et al., J. Phys. Chem. Lett. 22, 4573–4580 (2015) or Butkus et al., J. Chem. Phys. 140, 034306 (2014)] "vibronic" refers to the mixing of electronic and vibrational degrees of freedom. The term "excitonic-vibronic coupling" would then refer to a situation where vibronic states mix (again) with excitonic states. While the mentioned references in line 64 (ref. 5, 6, and 19–23) are well-suited examples to demonstrate the significance of vibronic effects, the term "excitonic-vibronic coupling" is not used in these publications. Closely connected to this issue is that "excitonic-vibronic coupling" as well as "electronic-vibronic coupling" are both used in the manuscript. The latter term is not properly defined even though it is part of the title and so it is not clear if the two terms (excitonic-vibronic coupling and electronic-vibronic coupling) are referring to the same situation. It would clarify the manuscript if the term "excitonic-vibronic coupling" is changed in accordance with literature and if only one (well defined) term is used throughout the manuscript.

Line 138: One of the major differences in the 2D spectra of the trimer and the monomer is the absence of the excited-state absorption at $\omega_1=16500$, $\omega_3=14750$ cm⁻¹ in the case of a trimer for waiting times of 400 fs and longer. What is the reason for this effect? Can the compensation of the positive ground-state bleach and stimulated emission signals with the negative excited-state absorption signal be a problem for the analysis of the monomer data?

Line 157: The authors state that in the case of a monomer only vibrational coherences can be observed. I think it would improve the clarity of the manuscript to state why only vibrational coherences are possible in the case of the monomer.

Line 181: The authors state that the "oscillations in the dynamical Stokes shift decay originate from the coherent coupling of vibrational modes to electronic states of a wavepacket". This statement is not straightforward to understand. Furthermore, I could not find a direct connection to ref. 40 which provides an excellent discussion of coherences in 2D spectroscopy but seems not to discuss the issue of dynamic Stokes shift explicitly. Could the authors elaborate more why these oscillations originate from coherent coupling of vibrational modes to electronic states, and provide more references since it is one of the central points of the manuscript?

Line 192: When referring to "gray lines", the authors should also refer to the corresponding figure number.

Line 222: The authors explain the observed signatures with the effect of phase synchronization. A publication from 2012 explored the origin of long-lived coherences in FMO using a vibronic exciton model [Christensson et al. J. Phys. Chem. B 116 (25), 7449-7454 (2012)] and the results were in good agreement with experimental work. The explanation of Christensson et al. for long-lived

coherences is based on "coherent superpositions of vibronic exciton states with dominant contributions from vibrational excitations on the same pigment". Could the author comment how their work relates to the work from Christensson et al.? Can the findings of Christensson et al. also explain (some) of the signatures in the present manuscript?

Line 297: The authors state that the asymmetric enhancement of specific peaks in the 2D spectrum fails to be a good indicator of "resonant excitonic-vibronic mixing". Could the authors provide possible reasons why this indicator is not sufficient in this case?

Line 338: I might have missed it, but I did not find a statement in the method section (or elsewhere in the manuscript) about the used polarization in the 2DES experiments. The authors might want to add this information for completeness.

Line 342: I think the pulse length should be stated as " $\sim 8\text{fs}$ " and not " $\sim 10\text{fs}$ " in accordance with line 68 in the supplementary information.

Comments to the supplementary material:

Line 149: The authors state that the second and third lifetime might include exciton-exciton annihilation. Can annihilation as a higher order process not be ruled out as a possible explanation since the authors state in line 81 that they measured in the linear response region?

Reviewer #2 (Remarks to the Author):

Reviewer #3 (Remarks to the Author):

In this paper the authors claim to provide the experimental verification of the quantum phase synchronization effect, theoretically proposed to justify long-living coherences in photosynthetic antennae.

Overall I believe that the experimental data do not convincingly support this claim. Also, I think that the description of the phenomena is not rigorous enough, especially considering the standards of Nature Comm. All considered, I do not recommend publication.

1) The first concern is about data analysis and interpretation.

a) While I do agree that the 2DES responses of the trimer and the monomer are different, I do not fully see that 'two diagonal peaks were observed for the trimer, with positive values corresponding to the bleaching/emission of the upper and lower excitonic levels (Fig. 2a), while only a single elongated diagonal peak was observed for the monomer (Fig. 2b).' (lines 139-141). I see in both spectra only an elongated single peak. Also, I do not see clearly the cross peak pinpointed by the pink square. A better analysis of the peak shape should be done to convince the reader of the effective presence of these peaks.

b) Let's, for now, overcome the previous point and assume that, although not clearly distinguishable, the diagonal and off-diagonal positions pinpointed by the authors identify in any case relevant coordinates. There are several examples in the literature where the formation of dimers reduces the coupling with vibrational degrees of freedom. This is manifested also here where the amplitude of beatings is considerably smaller in the trimer than in the monomer (figure 2c). Basically, there are no beatings in the trimer's signal and this means that the coupling of the

electronic transitions with the vibrational modes is negligible. This has been observed and commented several times in excitonic systems.

This is in clear contrast with the main conclusion of the paper, stating that the vibronic coupling is such to cause quantum phase synchronization and long-living beatings in the trimer. It is clear to me that this claim is based on the analysis of the beatings of the Stokes shift, not of the beatings of the 2DES signal. However, even if the observables are different, the physical origin at the base of the two phenomena is the same (i.e. vibronic coupling).

Also, looking carefully at the Fourier spectra in figure 3, I doubt that the authors can extract quantitative information (for example the claim that in the trimer, a few specific modes have a 50% higher intensity). The noise level is very high, and I do not think this allows for a quantitative claim.

c) It is not clear (and the description of the model on lines 221- 270 does not help too much...) what the connection is between the Stokes shift and the quantum phase synchronization. In other words, it is not clear why the presence of beatings should be considered a signature of quantum phase synchronization.

Overall, I think that the conclusions are not reliably supported by the data. Rather than invoking exotic quantum synchronization effects, I believe that the experimental data could be more reliably and convincingly justified with a much simpler interpretation:

During the delay time t_2 , the system undergoes relaxation processes through the vibronic pathways. As the authors wrote in the paper, such a vibrational cooling causes the dynamic Stokes shift and is manifested as the shifting of peaks down in the energy scale along the emission axis as a function of t_2 . The ground state bleaching and stimulated emission features (originating the peaks in the 2D maps) can, in principle, experience dynamic Stokes shifts on different timescales and with different vibronic coupling. This can cause some overlaps of positive and negative contributions, giving a complex time-dependence of both diagonal peaks and cross-peaks. Could this simple interpretation be ruled out?

2) The second problem is the accuracy of the language and the quality of the descriptions.

a) When describing the coupling between electronic and vibrational degrees of freedom, several times in the paper, including in the title, the authors mention 'electronic-vibronic' or 'excitonic-vibronic' coupling. This is inaccurate (not to say wrong). The term 'vibronic' already includes a mixing between electronic and vibrational degrees of freedom. Therefore 'excitonic-vibronic' coupling does not have any meaning: it is either an 'electronic-vibrational' or a 'vibronic' coupling. (see Fisher book). I suggest reconsidering the description in lines 128-134.

b) About the definition of an excitonic dimer. The formation of excitonic states intrinsically requires the presence of a strong coupling (see the seminal work by Kasha), which seems to be not the case here.

c) The description of the 2DES (lines 99-110) is a bit too hasty. I suggest fully rewriting this part with better accuracy. Also the sentence on lines 110-112 should be explained better. The connection between experimentally detected Stokes shift, protection of coherence, and quantum phase synchronization is absolutely unclear. The authors took for granted that these three concepts are related but the reason why it is so is not emerging from the paper.

Point-to-point replies to the reviewers' comments

All the questions are in red while the answers are in blue.

Reviewers' comments:

Reviewer #1 (Remarks to the Author):

The authors address the question how long-lasting coherences can survive in the presence of a noisy environment as it is the case for most biological systems. Using two-dimensional spectroscopy as well as other techniques, they probe the coherences in allophycocaynin trimers and monomers. The authors explain the differences of the monomer and trimer measurements with the help of a vibronic dimer model and identify two spectroscopic signatures of phase synchronization. The authors conclude that phase synchronization is a mechanism that explains long-lasting coherences in biological systems.

Vibronic coherences in photosynthetic antenna complexes are an ongoing research topic and thus the study is of fundamental scientific significance. The analysis is in principle convincing, and I recommend this work for publication in Nature Communications. However, I have a few issues that need to be addressed prior to publication.

Thanks to the reviewer for the lucid summary and recommendation of our work.

Comments to the main manuscript:

Q1: Line 63: The term “excitonic-vibronic coupling” is defined as the “...exchange of energy between excitonic and vibrational degrees of freedom”. The term “excitonic-vibronic coupling” is in my opinion confusing since in literature [compare, for example, ref. 19, Novelli et al., J. Phys. Chem. Lett. 22, 4573–4580 (2015) or Butkus et al., J. Chem. Phys. 140, 034306 (2014)] “vibronic” refers to the mixing of electronic and vibrational degrees of freedom. The term “excitonic-vibronic coupling” would then refer to a situation where vibronic states mix (again) with excitonic states. While the mentioned references in line 64 (ref. 5, 6, and 19–23) are well-suited examples to demonstrate the significance of vibronic effects, the term “excitonic-vibronic coupling” is not used in these publications. Closely connected to this issue is that “excitonic-vibronic coupling” as well as “electronic-vibronic coupling” are both used in the manuscript. The latter term is not properly defined even though it is part of the title and so it is not clear if the two terms (excitonic-vibronic coupling and electronic-vibronic coupling) are referring to the same situation. It would clarify the manuscript if the term “excitonic-vibronic coupling” is changed in accordance with literature and if only one (well defined) term is used throughout the manuscript.

A1: Thanks to the reviewer for the insightful questions. Frankly speaking, during the

writing of the manuscript, we were somewhat confused by the terminology scattered in the literature, especially in the earlier literature, the term “excitonic-vibronic coupling” has been used to describe excited states of the dimer with vibronic transition in its constituent monomers (Koch, C., et al., Journal of Luminescence 81 (1999); 171-181; Esser, B. et al. Journal of Luminescence 768~77 (1998) 530-533; Capek, V. , Eur. Phys. J. B 2, 37 – 49 (1998)). Later, “vibronic-exciton” is used to describe the excitons of dimer or aggregates, where vibronic excitation in the monomer is involved (Caycedo-Soler, F. et al., J. Phys. Chem. Lett. 2018, 9, 3446–3453). Especially when the vibrational frequency is in resonance with the electronic splitting of the exciton, it is called as the “exciton-vibronic resonance” (Butkus, V. et al., J. Chem. Phys. 140, 034306, 2014), while Polyutov et al., used “exciton-vibrational coupling” in the investigation of the excitonic dimer (Polyutov et al, Chem. Phys. 394,2128,2012). Therefore, we replace the term “ excitonic-vibronic coupling” with “exciton-vibrational coupling” in the revised version to be in accordance with most of the later literature. In our previous manuscript, we mixed the two terms, i.e., “excitonic-vibronic coupling” and “electronic-vibronic coupling”, and only one term “exciton-vibrational coupling” is used in the revised version.

Q2: Line 138: One of the major differences in the 2D spectra of the trimer and the monomer is the absence of the excited-state absorption at $\omega_1=16500$, $\omega_3=14750$ cm^{-1} in the case of a trimer for waiting times of 400 fs and longer. What is the reason for this effect?

A2: The peaks below the diagonal with negative amplitude are the excited-state absorption peaks. For the monomer, its central peak position is located at $\omega_1=16500$, $\omega_3=14750$ cm^{-1} ; while for the trimer, the two peaks at $\omega_1=15300$, $\omega_3=14750$ cm^{-1} ; and $\omega_1=16000$, $\omega_3=14750$ cm^{-1} respectively for the absorption of the two excitonic states. The difference between the excited-state absorption signals is that the lifetime for the excited state of the monomer is much longer with a much larger initial amplitude than that of the trimer, which leads to that at 400 fs delay after the excitation, the excited-state absorption signal for the trimer in 2D map disappears quickly, while that of the monomer is still intense enough (Figure R1a or Figure S13a).

Q3: Can the compensation of the positive ground-state bleach and stimulated emission signals with the negative excited-state absorption signal be a problem for the analysis of the monomer data?

A3: The compensation with the negative excited-state absorption signal would not be a problem for the analysis of the monomer data. The reasons are as follows: for the decay kinetics, the excited-state absorption (ESA) and the ground-state bleaching/stimulated emission (SE) peaks are well separated. As to the dynamical Stokes shift, the ESA signal (negative) and the SE signal (positive) are all red-shifted as shown in Figure R1b or Figure S13b in the revised version, which clearly shows that the spectral shift in the negative ESA signal would not compensate with that of the SE,

while the ground-state bleaching signal does not change with time.

Figure R1. Kinetic analysis of ESA signals. (a) The kinetics of ESA signal for the monomer (orange) and trimer (green and blue). The three-exponential fitting results are superimposed on top of the original kinetics. The kinetics of trimers were multiplied by a factor of 3 for ease of observation and comparison. (b) Peak shift traces of the negative peak (ESA) and the positive peak (SE) at $\omega_1=16000\text{ cm}^{-1}$. Both traces show almost the same red-shift process.

Q4: Line 157: The authors state that in the case of a monomer only vibrational coherences can be observed. I think it would improve the clarity of the manuscript to state why only vibrational coherences are possible in the case of the monomer.

A4: The rAPC monomer contains two pigments, which are spatially separated by a distance of 50 angstroms. Consequently, the intermolecular coupling between these pigments is negligible. As a result, the coherent oscillating signal observed in the monomer can solely be attributed to intramolecular vibrational contributions. This has been clarified at the end of page 7 in the revised version.

Q5: Line 181: The authors state that the “oscillations in the dynamical Stokes shift decay originate from the coherent coupling of vibrational modes to electronic states of a wavepacket”. This statement is not straightforward to understand.

A5: Line 181 reads “These oscillations in the dynamical Stokes shift decay curve originate from the coherent coupling of vibrational modes to electronic states as a wavepacket”.

Taking the monomer as an example, when vibronic transition occurs, except for the 0-0 electronic transition, other vibrational modes are also excited with a transition probability proportional to their respective Huang-Rhys factors. In an impulsive excitation limit, these coherently excited vibrational modes in the excited state constitute a vibrational wavepacket. Since all the individual modes are vibronically excited, this wavepacket is coupled to the excited electronic state. When further considering that the vibrational modes are underdamped, the effect of the vibrational modes on the electronic energy level can be treated by a multimode Brownian oscillator model. As a result, the time-dependent energy level of the excited state

$E_e(t)$ with respect to the nuclear motion of j th normal mode can be expressed as follow,

$$E_e(t) \equiv \omega_{eg} - \sum_j [\lambda_j - 2\lambda_j M_j(t)]$$

where ω_{eg} is the 0-0 transition electronic transition energy; λ_j is the reorganization energy of the j th mode, and $M_j(t)$ is the correlation function reflecting the motion of the j th mode under solvent friction with γ_j as the relaxation constant

$$M_j(t) = e^{-\frac{\gamma_j t}{2}} \left(\cos \bar{\omega}_j t + \frac{\gamma_j}{2\bar{\omega}_j} \sin \bar{\omega}_j t \right)$$

$$\bar{\omega}_j = [\omega_j^2 - (\gamma_j/2)^2]^{1/2}$$

for the underdamped oscillator, especially when $\bar{\omega}_j \gg \gamma_j$, $\bar{\omega}_j = \omega_j$

$$E_e(t) \equiv \omega_{eg} - \sum_j \left\{ \lambda_j - 2\lambda_j e^{-\frac{\gamma_j t}{2}} \left(\cos \omega_j t + \frac{\gamma_j}{2\bar{\omega}_j} \sin \omega_j t \right) \right\}$$

Obviously, the dynamical Stokes shift is modulated by a motion of the vibrational wavepacket reflected in the second term on the right side of the above equation.

We added a paragraph at the bottom of page 8 in the revised version to clarify this point:

“In a 2DES experiment, the system is excited with ultrafast laser pulses. This impulsive laser excitation of vibrational modes to the Franck-Condon region in the molecular excited state produces wavepackets in the excited state and “holes” in the ground state. As time evolves, the excited state begins to relax through intramolecular and intermolecular vibrational relaxation and solvent reorganization. Relaxation of the excited electronic state manifests as a shift in frequency along the emission frequency of the 2DES spectrum as the energy gap between the ground and excited state decreases. This energy relaxation is referred to as the dynamic Stokes shift and can be described by the Stokes shift function $S(t)$, which exhibits oscillatory behavior, and the oscillatory components are assigned to intramolecular vibrational modes. Fourier transformation of $S(t)$ gives rise to the coherence spectrum of dynamical Stokes shift. Therefore, the dynamic Stokes shift is a common means for characterizing ultrafast solvation dynamics of electronically excited states, hence the energy

dissipation to the environments as well as the intramolecular or solvent vibrational modes participating in the energy dissipation^{34, 43, 44}.”

Q6: Furthermore, I could not find a direct connection to ref. 40 (Yue, wavepacket) which provides an excellent discussion of coherences in 2D spectroscopy but seems not to discuss the issue of dynamic Stokes shift explicitly. Could the authors elaborate more why these oscillations originate from coherent coupling of vibrational modes to electronic states, and provide more references since it is one of the central points of the manuscript?

A6: Reference 40 cited here is to give support for the vibrational wavepacket consisting of multiple vibrational modes in the excited state, and this reference is not related to the dynamical Stokes shift. More related references are added in the revised version.

Q7: Line 192: When referring to “gray lines”, the authors should also refer to the corresponding figure number.

A7: We changed to “grey lines in Figure 4” in the revised version.

Q8: Line 222: The authors explain the observed signatures with the effect of phase synchronization. A publication from 2012 explored the origin of long-lived coherences in FMO using a vibronic exciton model [Christensson et al. J. Phys. Chem. B 116 (25), 7449-7454 (2012)] and the results were in good agreement with experimental work. The explanation of Christensson et al. for long-lived coherences is based on “coherent superpositions of vibronic exciton states with dominant contributions from vibrational excitations on the same pigment“. Could the author comment how their work relates to the work from Christensson et al.? Can the findings of Christensson et al. also explain (some) of the signatures in the present manuscript?

A8: It has been agreed that pure electronic coherence dephasing time is around 100 fs. In order to account for the long coherence time exceeding that of pure electronic coherence, several models have introduced certain kinds of correlations, e.g., static or dynamic correlations in the site energies of the pigments, and some authors suggested that this can be realized by the motion of the protein matrix where pigments are embedded. Coherence with ps dephasing time in the electronic model thus requires a correlation of both dynamic fluctuations and static distributions of the transition energies of different pigments.

Christensson *et al.*, demonstrated theoretically that a biphasic decay feature exists in the vibronic exciton model, explicitly, the oscillations show a biphasic behavior, where the initial 200 fs decay of the oscillation is due to the decay of coherences between vibronic exciton states localized on different pigments, while the long-lived

oscillations reflect coherences between vibronic exciton states localized on the same pigment (“intrapigment”) due to strong exciton-vibrational coupling at the near-resonant condition (excitonic energy difference: $\Delta E=217\text{ cm}^{-1}$, vibrational frequency $\omega=185\text{ cm}^{-1}$), where the excitonic state contain 68% of the vibrational mode, this leads to that the two exciton levels will experience highly correlated fluctuations. This type of correlation leads to slow homogeneous dephasing of coherences with large “intrapigment” characters. Apparently, Christensson’s model describes the type of long-lived oscillation with strong exciton-vibrational coupling.

In APC trimer, dephasing of the vibronic-exciton is also biphasic, the fast decay is from pure electronic coherence which has been experimentally determined as 10-30 fs (Edington, M. D., Riter, R. E., Beck, W. F., J. Phys. Chem., 99,15699-15704,1995). While the slow decay phase has a distinct vibronic feature, i.e., the wavefunction is a mixture of the electron and nuclear motion through Franck-Condon transition. The

electronic coupling strength $\left| \frac{V}{\Delta E} \right| = 0.42$ is rather weak, where ΔE is the disorder

of the site energies, and V is the excitonic coupling matrix element. Furthermore, the Huang-Rhys factors for the vibrational modes larger than 490 cm^{-1} for the APC trimer are small (less than 0.1) (J. M. Zhang, Y. J. Shiu, M. Hayashi, K. K. Liang, C. H. Chang, V. Gulbinas, C. M. Yang, T.-S. Yang, H. Z. Wang, Yit-Tsong Chen and S. H. Lin, J. Phys. Chem. A 2001, 105, 8878-8891), both the facts suggest that APC trimer is a weak electronic and excitonic-vibrational coupling system.

The difference lies in that owing to the large site energy in the APC trimer, the two exciton levels are located almost on different pigments, thus they can not be considered as “intrapigment”. However, once the anti-symmetric collective modes have been dissipated through electronic dephasing owing to their coupling to the excitonic levels, only the total symmetric collective modes remain, and this leads to the correlated energy fluctuation of the two excitonic levels located on two different pigments. From this point of view, the interacted pigment pair behaves as a single molecule, having the “intrapigment” feature. Unlike the Christensson’s vibronic exciton model for the vibronic exciton of strong coupling, the seemingly “intrapigment” feature is not an inherent property for the APC trimer of rather weaker coupling, but it is an acquired property by phase synchronization, and this process takes some period of time. Therefore, the findings of Christensson *et al.* are invalid to the current system.

We added a discussion paragraph including the above arguments on pages 16-17.

Q9: Line 297: The authors state that the asymmetric enhancement of specific peaks in the 2D spectrum fails to be a good indicator of “resonant excitonic-vibronic mixing”. Could the authors provide possible reasons why this indicator is not sufficient in this case?

A9: We have observed the asymmetric enhancement of specific peaks in the 2D

spectrum also in most of pure vibrational modes in the monomer where there is no excitonic feature. This is clarified at the bottom of page 7 in the revised version.

Q10: Line 338: I might have missed it, but I did not find a statement in the method section (or elsewhere in the manuscript) about the used polarization in the 2DES experiments. The authors might want to add this information for completeness.

A10: The pulses in all the ultrafast spectroscopic experiments are parallel-polarization, which has been described in the Method section of the revised manuscript.

Q11: Line 342: I think the pulse length should be stated as “~8fs” and not “~10fs” in accordance with line 68 in the supplementary information.

A11: Thanks for this suggestion. We have modified the statement about the pulse width from “~10 fs” to “~8 fs”.

Comments to the supplementary material:

Q12: Line 149: The authors state that the second and third lifetime might include exciton-exciton annihilation. Can annihilation as a higher order process not be ruled out as a possible explanation since the authors state in line 81 that they measured in the linear response region?

A12: In global analysis of the spectral evolution, we speculated that the second and third lifetime constants may contain two contributions based on the similarity of spectral shapes: one is the energy relaxation process to the ground state and the other is the exciton-exciton annihilation process (higher order process). Even when measured in the linear response range, non-negligible higher-order signals are common in 2D spectroscopic measurements of other light-harvesting antenna systems, such as LHCII (*J. Phys. Chem. Lett.* 8, 257-263 (2017) ; *J. Chem. Phys.* 156, 125101 (2022)). However, it is worth noting that the second and third lifetime constants of 1.6 and 57 ps, respectively, are much longer than the observed coherent oscillation discussed in the text (less than 1 ps, as shown in Fig. 2c). Thus, they do not correspond to the coherent phenomena discussed on a time scale. Consequently, we can rule out exciton-exciton annihilation processes that occur on slower time scales as a possible explanation for the observed coherence phenomenon.

Reviewer #2 (Remarks to the Author):

Thanks to the reviewer for the effort in reviewing our manuscript.

Reviewer #3 (Remarks to the Author):

In this paper the authors claim to provide the experimental verification of the quantum phase synchronization effect, theoretically proposed to justify long-living coherences in photosynthetic antennae.

Overall I believe that the experimental data do not convincingly support this claim. Also, I think that the description of the phenomena is not rigorous enough, especially considering the standards of Nature Comm. All considered, I do not recommend publication.

Thanks to the reviewer for critical reading and very insightful suggestions and comments.

1) The first concern is about data analysis and interpretation.

Q1: a) While I do agree that the 2DES responses of the trimer and the monomer are different, I do not fully see that ‘two diagonal peaks were observed for the trimer, with positive values corresponding to the bleaching/emission of the upper and lower excitonic levels (Fig. 2a), while only a single elongated diagonal peak was observed for the monomer (Fig. 2b).’ (lines 139-141). I see in both spectra only an elongated single peak. Also, I do not see clearly the cross peak pinpointed by the pink square. A better analysis of the peak shape should be done to convince the reader of the effective presence of these peaks.

A1: Thanks to the reviewer for pointing out the unobvious “two diagonal peaks” for the trimer. In our assignment, the two diagonal peaks are located at (16000, 16000 cm^{-1}) and (15300, 15300 cm^{-1}), respectively.

We adjusted the contour level in Figure 2a (from 11% to 8% of the maximum value) to make the upper exciton level signal at 16000 cm^{-1} on the diagonal more pronounced. The pink square was also repositioned to make the indicated cross peaks at (16000, 15300) cm^{-1} more visible (Figure 2a in the revised version). In addition, we also plot the spectra along the diagonal at different waiting times. Through the normalized spectra, we can more clearly observe the diagonal peak signal at 16000 cm^{-1} as shown by the arrow in Figure R2 (Figure S10 in the revised version).

Figure R1. The spectra along the diagonal obtained at different waiting times.

We believe that the low distinction between these two diagonal peaks is due to the large inhomogeneous broadenings at room temperature, as shown in the steady-state spectra of the rAPC trimer (Fig. 1b). Therefore, we verify this idea by performing additional 2DES experiments at 77K to reduce the effect of inhomogeneous broadening. As shown in Figure R3 (Figure S11 in the revised version), both diagonal peaks and the cross peak below the diagonal which corresponds to the energy transfer process can be clearly observed at low temperatures, while the two absorption peaks above the diagonal are also observed which can be assigned to the transitions of the excitonic levels to the upper levels of the excited states.

Figure R3. Absorptive 2D spectra of the rAPC trimer at 77 K.

Q2: Let's, for now, overcome the previous point and assume that, although not clearly

distinguishable, the diagonal and off-diagonal positions pinpointed by the authors identify in any case relevant coordinates.

A2: We make more clearly the pinpointed off-diagonal position by the 2DES measurement at 77 K as shown in Figure R3 and Figure S11 in the revised version.

Q3: There are several examples in the literature where the formation of dimers reduces the coupling with vibrational degrees of freedom.

A3: Yes, we noted that there are some reports showing that the formation of dimers reduces the coupling with vibrational degrees of freedom in the photosynthetic complexes. However, only the low-frequencies in a region below 300 cm^{-1} are strongly suppressed by exciton-vibronic delocalization, while the high-frequency modes are likely to be unaffected (Mike Reppert, *J. Phys. Chem. B* 2020, 124, 10024–10033). The same phenomenon has also been reported in the LH2, a light-harvesting complex from the photosynthetic bacteria (Shawn Irgen-Gioro, Karthik Gururangan, Austin P. Spencer, and Elad Harel, *J. Phys. Chem. Lett.* 2020, 11, 10388–10395). As to the current case, the observed reduced intensities of the vibrational modes between 600-820 cm^{-1} can not be caused by the electronic delocalization in the dimer as reported.

This has been addressed in the revised version on the bottom lines of page 11.

Q4: This is manifested also here where the amplitude of beatings is considerably smaller in the trimer than in the monomer (figure 2c). Basically, there are no beatings in the trimer's signal and this means that the coupling of the electronic transitions with the vibrational modes is negligible (Huang Ris). This has been observed and commented several times in excitonic systems. This is in clear contrast with the main conclusion of the paper, stating that the vibronic coupling is such to cause quantum phase synchronization and long-living beatings in the trimer.

A4: The beating signals in APC trimer has not only been observed by our 2DES, but also been observed by transient absorption spectroscopy, with similar vibrational frequencies (J. M. Zhang, Y. J. Shiu, M. Hayashi, K. K. Liang, C. H. Chang, V. Gulbinas, C. M. Yang, T.-S. Yang, H. Z. Wang, Yit-Tsong Chen and S. H. Lin, *J. Phys. Chem. A* 2001, 105, 8878-8891).

It has been accepted that the Huang-Rhys factor of a vibrational mode S^ω in a pigment would be different when the pigment molecules form excitonic dimer or J -aggregates. For multichromophoric systems with J -coupling, the mixing of excited electronic states also has the potential to affect the coupling strength of vibrations because the resulting excitonic states are formed from combinations of pigment vibronic states. The change in a particular vibrations's coupling strength to the excitonic manifold due J -coupling can be described with a generalized Huang-Rhys factor (Shawn Irgen-Gioro, Karthik Gururangan, Austin P. Spencer, and Elad Harel, *J.*

Phys. Chem. Lett. 2020, 11, 10388–10395). Under exciton-vibrational resonance and non-adiabatic conditions, the excitonic and vibrational degrees of freedom become strongly mixed and vibronic transitions gain intensity by redistribution of the total oscillator strength. For simplicity, the generalized Huang-Rhys factor S_g^ω can be

expressed as $S_g^\omega \propto \frac{S^\omega}{\Delta}$, where Δ is the magnitude of the detuning between the

excitonic energy splitting and the vibrational frequency ω . Under the idealized resonance condition, $\Delta=0$ (Maxim F. Gelin, , Lipeng Chen, Raffaele Borrelli,, Erling, Thyraug, *Chemical Physics* 528 (2020) 110495). Accordingly, the resonance modes would have a substantially enhanced generalized Huang-Rhys factor with respect to the vibrational mode in the monomer. In the excitonic system as B850 ring in LH2 protein-pigment complex consisting of 9 pairs of strongly coupled bacteriochlorophyll (BChl) molecules as in the antenna of photosynthetic bacteria, it has been shown that the strong vibronic coupling due to excitonic wave function overlap, the vibrational mode in resonance with the excitonic energy gap would have an enhanced Huang-Rhys factor 6 times larger in the magnitude than that of the individual pigment. (Caycedo-Soler, F. ,et al., *J. Phys. Chem. Lett.* 2018, 9, 3446–3453). The resonance-selected enhancement was also indicated in a publication, where the excitonic splitting energy in B850 ring is 680 cm^{-1} . The bacteriochlorophyll *a* has two vibrational modes of 730 and 570 cm^{-1} of equal intensity. Under the non-resonance condition such as B820, the probed oscillation amplitudes of these two mods are also of the same intensity, while at the near resonance condition as in B850, the amplitude of 730 cm^{-1} mode (intensity: 17, a.u.) is obviously larger than that of the 570 cm^{-1} (13, a.u.). (Shawn Irgen-Gioro, Karthik Gururangan, Austin P. Spencer, and Elad Harel, *J. Phys. Chem. Lett.* 2020, 11, 10388–10395). Therefore, the observed smaller beating intensity can not be attributed to the suggested negligible Huang-Rhys factor in the excitonic dimer.

Q5: It is clear to me that this claim is based on the analysis of the beatings of the Stokes shift, not of the beatings of the 2DES signal. However, even if the observables are different, the physical origin at the base of the two phenomena is the same (i.e. vibronic coupling).

A5: Our claim is based on both the beatings of the 2DES signal and of the oscillations in the dynamical Stokes shift. The former gives a 50% intensity loss in the beating amplitudes of the resonant modes, while these resonant modes would disappear completely in the dynamical Stokes shift. Both the experimental results and the theoretical predictions agree well with each other. Both of these two features reflect two aspects of the quantum phase synchronization in an excitonic dimer of weak coupling.

Q6: Also, looking carefully at the Fourier spectra in figure 3, I doubt that the authors

can extract quantitative information (for example the claim that in the trimer, a few specific modes have a 50% higher intensity). The noise level is very high, and I do not think this allows for a quantitative claim.

A6: In the revised version, we acquired the coherence spectra for the α - and β -units respectively in addition to that of monomer with three different methods, i.e., 2DES, HDTG and BBTA as shown in Figure S17 in the revised version. Total nine sets of data give an averaged amplitude reduction ratio of $49\% \pm 7\%$ for the trimer over those of the monomer and the two respective subunits at the two resonant modes of 660 and 805 cm^{-1} (Table 1 in the revised version). We also replaced the coherence spectra of monomer with those of α -subunit of better signal-to-noise ratio as shown in Figure 4 in the revised version. As to the dynamical Stokes shift, the background noise level of 2DES(BBTA)-detected coherence spectrum was about 12% (4%) of the FT spectrum maximum (Figure 4 in the revised version).

Q7: It is not clear (and the description of the model on lines 221- 270 does not help too much...) what the connection is between the Stokes shift and the quantum phase synchronization. In other words, it is not clear why the presence of beatings should be considered a signature of quantum phase synchronization.

A7: We put a summarizing paragraph to account for how quantum phase synchronization could lead to the disappearance of the resonant modes in the dynamical Stokes shift prior to the theoretical derivation in the first paragraph of page 13:

“This synchronization idea was inspired by the classical analogue known as Huygens’ clock (Fig. 5a), where the two pendulums are coupled. The in-phase collective motion is more energy dissipative than the anti-phase one, resulting in the anti-synchronized motion of the pendulum clocks⁵¹. If this synchronization mechanism also presents in the exciton-vibrational coupled dimer, it can be expected that phase synchronization of the resonant collective modes would lead to these modes becoming less energy dissipative as indicated by our experimental observation in Fig. 3”

Q8: Overall, I think that the conclusions are not reliably supported by the data. Rather than invoking exotic quantum synchronization effects, I believe that the experimental data could be more reliably and convincingly justified with a much simpler interpretation:

During the delay time t_2 , the system undergoes relaxation processes through the vibronic pathways. As the authors wrote in the paper, such a vibrational cooling causes the dynamic Stokes shift and is manifested as the shifting of peaks down in the energy scale along the emission axis as a function of t_2 . The ground state bleaching and stimulated emission features (originating the peaks in the 2D maps) can, in principle, experience dynamic Stokes shifts on different timescales and with different

vibronic coupling. This can cause some overlaps of positive and negative contributions, giving a complex time-dependence of both diagonal peaks and cross-peaks. Could this simple interpretation be ruled out?

A8: The ground-state bleaching spectrum is a mirror image of the steady-state absorption spectrum with an intensity of opposite sign and it would not change with time. However, the interaction of the excited-state potential surface with the solvents leads to the stimulated emission spectrum changing with time. Therefore dynamical Stokes shift traces the time-dependent stimulated emission spectra, which is directly related to the nuclear motion of the underdamped modes as shown in Fig. R4, while the related physical process has been well studied theoretically in detail (Yi Jing Yan and Shaul Mukamel, *Phys. Rev. A.*, 1990, 41(11), 6458-6504). Therefore the possibility that “the ground state bleaching and stimulated emission cause some overlaps of positive and negative contributions, giving a complex time-dependence of both diagonal peaks and cross-peaks” can be ruled out.

Figure R4. Schematic diagram illustrating the dynamical Stokes shift for a strongly overdamped mode U , in the spectral diffusion limit (From Yan et al., *Phys. Rev. A.* 1990, 41, 6485). When underdamped modes are considered, the time-dependent dynamical Stokes shift would be modulated by the nuclear motion of the mode.

2) The second problem is the accuracy of the language and the quality of the descriptions.

Q9: a) When describing the coupling between electronic and vibrational degrees of freedom, several times in the paper, including in the title, the authors mention ‘electronic-vibronic’ or ‘excitonic-vibronic’ coupling. This is inaccurate (not to say wrong). The term ‘vibronic’ already includes a mixing between electronic and vibrational degrees of freedom. Therefore ‘excitonic-vibronic’ coupling does not have any meaning: it is either an ‘electronic-vibrational’ or a ‘vibronic’ coupling. (see

Fisher book). I suggest reconsidering the description in lines 128-134.

A9: Thanks to the reviewer for the insightful suggestions, we fully agree with the reviewer's comments. This concern is the same as that raised by the other reviewer (Answer 1 to the reviewer 1). we replace the terms “excitonic-vibronic coupling” and “electronic-vibrational” with “exciton-vibrational coupling” in the revised version to be in accordance with most of the later literature (Polyutov et al, Chem. Phys. 394,2128,2012).

Q10: b) About the definition of an excitonic dimer. The formation of excitonic states intrinsically requires the presence of a strong coupling (see the seminal work by Kasha), which seems to be not the case here.

A10: Currently, the quantitative criterion for characterization of the electronic coupling strength in the excitonic dimer is as follows:

By comparing the magnitudes of the disorder of the site energies (ΔE) with the excitonic coupling matrix element V ($V=2|J|$), the excitons can be classified into

three cases: the regime of strongly coupled excitonic dimer with the limit $\left| \frac{V}{\Delta E} \right| \gg 1$;

the regime of weak coupling, $\left| \frac{V}{\Delta E} \right| \ll 1$. Typical values are 20 for strong coupling; 2

for intermediate coupling, and 0.05 for weak coupling (Cogdell R., et al., Quarterly Reviews of Biophysics 39, 3 (2006); Rohr, M. I. S., et al., J. Phys. Chem. C 2018, 122,

8082–8093). In our case, $\Delta E = 800 \text{ cm}^{-1}$, $J=155 \text{ cm}^{-1}$, and $\left| \frac{V}{\Delta E} \right| = 0.39$, which is in

between intermediate and weak coupling. In the literature, the coupling constants for the pigments in photosynthetic antennas are in a range around 250 to 10 cm^{-1} (Friedl, C., Phys. Chem. Chem. Phys., 2022, 24, 5014). Considering the varied site energies, the coupling strengths are also different.

The definition of the coupling strength has been addressed in the discussion paragraph at the beginning of page 17 in the revised version.

Q11: c) The description of the 2DES (lines 99-110) is a bit too hasty. I suggest fully rewriting this part with better accuracy. Also the sentence on lines 110-112 should be explained better. The connection between experimentally detected Stokes shift, protection of coherence, and quantum phase synchronization is absolutely unclear. The authors took for granted that these three concepts are related but the reason why it is so is not emerging from the paper.

A11: Thanks to the reviewer for the suggestion, we have rewritten this part

according to the reviewer's suggestion on pages 4-5 in the revised version.

REVIEWERS' COMMENTS

Reviewer #1 (Remarks to the Author):

The authors expanded their explanation and performed additional experiments. Most of my comments are satisfactorily answered and the manuscript can in principle be published in Nature Communications.

I have two follow-up remarks regarding the response of the authors.

A9: I might have formulated my first question a bit imprecise. I understand that the authors cannot use the asymmetric enhancement of specific peaks in the 2D spectrum as an indicator of exciton-vibrational coupling since they observe this behavior both in the monomer and the trimer. Do the authors know why this feature is present in the monomer and the trimer, i.e., what is the physical reason for this observation?

A11: The authors state that "in the linear response range, non-negligible higher-order signals are common in 2D spectroscopic measurement of other light-harvesting complexes." I disagree with this statement. In other words, if measurements were indeed performed in the linear regime, exciton-exciton annihilation could not contribute to signal.

To explain this, let us consider how different orders of nonlinearity contribute to an observed signal. The third-order response scales linearly with the excitation intensity. Any higher-order response has a different power dependence [P. Hamm, M. Zanni, Concepts and Methods of 2D Infrared Spectroscopy, or J. Lüttig et al., J. Phys. Chem. Lett. 2023, 14, 7556]. For example, exciton-exciton annihilation can only be observed if at least a fifth-order response (that scales quadratically with the excitation intensity) contributes to the signal. In this sense, the observation of exciton-exciton annihilation and measuring in the linear range are contradictory statements. A possible explanation for the observation of annihilation despite measuring in the linear regime might be that when measuring the linearity, the deviation is small and not visible if only a few intensities are measured. Also, the intensity might fluctuate leading to some error. However, I would consider this a problem of the measurement. An alternative approach to quantify the degree of contribution from higher-order signals is to measure at different excitation intensities and check if the fitted time constants remain the same for different intensities.

In any case, I agree with the statement that the time constants are much longer than the coherent oscillations discussed in this paper.

Reviewer #2 (Remarks to the Author):

Reviewer #3 (Remarks to the Author):

I must say that I am extremely satisfied with the responses provided by the authors. Initially, when I read the first version of the paper, I had my doubts and even recommended rejecting it. However, the authors did an excellent job of addressing my concerns and those of the other reviewer by providing clearer measurements and supporting evidence to back up their original claims. I also appreciated the addition of their theoretical descriptions and analysis. Therefore, I now recommend accepting the paper for publication.

Point-to-point replies to the reviewers' comments

All the questions are in red while the answers are in blue.

Reviewers' comments:

Reviewer #1 (Remarks to the Author):

The authors expanded their explanation and performed additional experiments. Most of my comments are satisfactorily answered and the manuscript can in principle be published in Nature Communications.

Thanks to the reviewer for the recommendation of our work.

I have two follow-up remarks regarding the response of the authors.

Q1: I might have formulated my first question a bit imprecise. I understand that the authors cannot use the asymmetric enhancement of specific peaks in the 2D spectrum as an indicator of exciton-vibrational coupling since they observe this behavior both in the monomer and the trimer. Do the authors know why this feature is present in the monomer and the trimer, i.e., what is the physical reason for this observation?

A1: We give a possible account for the asymmetric enhancement of anti-diagonal cross peaks in the 2D coherence map, i.e., the amplitude of the peak below the diagonal corresponding to the ground state is larger than that above the diagonal for the excited state. If only an excitonic dimer is considered, these two peaks would have equal amplitude. In contrast, if only vibrational coherence is considered in a displaced oscillator model, the amplitude for the ground state would be larger than that of the excited state based on their corresponding Feynman diagram [V. Butkus et al., Chem. Phys. Lett. 545, 40-43 (2012); Franco V. de A. Camargo et al. Phys. Rev. Lett. 118, 033001 (2017)]. Thus, we added a sentence on page 13 of the revised version as follows with a citation of the above two references:

“The reason for the amplitude enhancement at the cross peak below the diagonal in the rephasing coherence maps for both the trimer and monomer could be that, in the vibrational coherence map, the ground-state vibrational coherence is dominant over that of the excited-state as revealed by the corresponding Feynman diagrams.”

Q2: The authors state that “in the linear response range, non-negligible higher-order signals are common in 2D spectroscopic measurement of other light-harvesting complexes.” I disagree with this statement. In other words, if measurements were indeed performed in the linear regime, exciton-exciton annihilation could not contribute to signal.

To explain this, let us consider how different orders of nonlinearity contribute to an observed signal. The third-order response scales linearly with the excitation intensity.

Any higher-order response has a different power dependence [P. Hamm, M. Zanni, Concepts and Methods of 2D Infrared Spectroscopy, or J. Lüttig et al., J. Phys. Chem. Lett. 2023, 14, 7556]. For example, exciton-exciton annihilation can only be observed if at least a fifth-order response (that scales quadratically with the excitation intensity) contributes to the signal. In this sense, the observation of exciton-exciton annihilation and measuring in the linear range are contradictory statements. A possible explanation for the observation of annihilation despite measuring in the linear regime might be that when measuring the linearity, the deviation is small and not visible if only a few intensities are measured. Also, the intensity might fluctuate leading to some error. However, I would consider this a problem of the measurement. An alternative approach to quantify the degree of contribution from higher-order signals is to measure at different excitation intensities and check if the fitted time constants remain the same for different intensities.

In any case, I agree with the statement that the time constants are much longer than the coherent oscillations discussed in this paper.

A2: We agree that if measurements were indeed performed in the linear regime, exciton-exciton annihilation could not contribute to the signals since any high-order response has a different power dependence. Therefore, we have removed the possible explanation of the exciton-exciton annihilation process concerning the lifetime in the revised supporting information, which does not affect the main conclusions of the article.

Reviewer #2 (Remarks to the Author):

Reviewer #3 (Remarks to the Author):

I must say that I am extremely satisfied with the responses provided by the authors. Initially, when I read the first version of the paper, I had my doubts and even recommended rejecting it. However, the authors did an excellent job of addressing my concerns and those of the other reviewer by providing clearer measurements and supporting evidence to back up their original claims. I also appreciated the addition of their theoretical descriptions and analysis. Therefore, I now recommend accepting the paper for publication.

Thanks to the reviewer for his appreciation and recommendation of our work.